EMBO
Molecular Medicine

# Malaria inflammation by xanthine oxidase-produced reactive oxygen species

Maureen C Ty[1], Marisol Zuniga[1], Anton Götz[1], Sriti Kayal[2], Praveen K Sahu[3], Akshaya Mohanty[4], Sanjib Mohanty[3], Samuel C Wassmer[5] & Ana Rodriguez[1,*]

## Abstract

**Malaria is a highly inflammatory disease caused by *Plasmodium* infection of host erythrocytes. However, the parasite does not induce inflammatory cytokine responses in macrophages *in vitro* and the source of inflammation in patients remains unclear. Here, we identify oxidative stress, which is common in malaria, as an effective trigger of the inflammatory activation of macrophages. We observed that extracellular reactive oxygen species (ROS) produced by xanthine oxidase (XO), an enzyme upregulated during malaria, induce a strong inflammatory cytokine response in primary human monocyte-derived macrophages. In malaria patients, elevated plasma XO activity correlates with high levels of inflammatory cytokines and with the development of cerebral malaria. We found that incubation of macrophages with plasma from these patients can induce a XO-dependent inflammatory cytokine response, identifying a host factor as a trigger for inflammation in malaria. XO-produced ROS also increase the synthesis of pro-IL-1β, while the parasite activates caspase-1, providing the two necessary signals for the activation of the NLRP3 inflammasome. We propose that XO-produced ROS are a key factor for the trigger of inflammation during malaria.**

**Keywords** inflammation; malaria; NLRP3; oxidative stress; reactive oxygen species

**Subject Categories** Immunology; Microbiology, Virology & Host Pathogen Interaction

## Introduction

Malaria induces an extremely high inflammatory response which, coupled to the sequestration of *Plasmodium falciparum*-infected red blood cells (iRBC), plays a key role in the life-threatening pathologies associated with this disease (Clark, 2007; Clark *et al*, 2008). A remarkable paradox in malaria research is that the strong inflammatory cytokine response and high fevers in patients cannot be modeled *in vitro*, where incubation of the parasite with immune cells results in little to no response in immune cells, such as macrophages (Scragg *et al*, 1999; Couper *et al*, 2010; Zhou *et al*, 2012) or dendritic cells (Elliott *et al*, 2007; Giusti *et al*, 2011; Götz *et al*, 2017). It is important to note that older publications showing *in vitro* inflammation were latter attributed to mycoplasma contamination of the cultures (Rowe *et al*, 1998).

The levels of oxidative stress in the circulation of malaria patients are elevated, as it is evident by the presence of high concentrations of plasma malondialdehyde, a by-product of lipid peroxidation, that is indicative of elevated oxidative stress (Narsaria *et al*, 2012). There are several sources of oxidative stress described in malaria: *Plasmodium* growth in erythrocytes (Atamna & Ginsburg, 1993), the macrophage oxidative burst (Kharazmi *et al*, 1987), and the upregulation of oxidative enzymes in the host (Iwalokun *et al*, 2006). Although they can all generate reactive oxygen species (ROS) during malaria, the relative contributions of each during infection are not known.

Upregulation of xanthine oxidase (XO), a host oxidative enzyme, has been documented in the circulation of malaria patients, infected monkeys, and mice (Tubaro *et al*, 1980; Srivastava *et al*, 1992; Siddiqi *et al*, 1999; Iwalokun *et al*, 2006). The enzyme precursor of XO, xanthine dehydrogenase, is upregulated by activation of type I IFN receptor very early upon *Plasmodium* infection in mice (Guermonprez *et al*, 2013), suggesting that early sensing of malaria results in increased oxidative stress.

In general, oxidative stress is known to cause inflammation through the destruction of tissues and release of danger signals by necrotic cells (Mittal *et al*, 2014). Here, we show an alternative mechanism where oxidative stress directly activates human macrophages to release inflammatory cytokines, identifying extracellular ROS a potent trigger of inflammation. In the context of malaria, we found that an inflammatory cytokine response could be triggered in macrophages simply by incubation with the plasma of malaria patients and that this response is dependent on XO, since a specific inhibitor of this enzyme can inhibit this response. These results

1 Department of Microbiology, New York University School of Medicine, New York, NY, USA
2 Department of Biotechnology and Medical Engineering, National Institute of Technology, Rourkela, Odisha, India
3 Center for the Study of Complex Malaria in India, Ispat General Hospital, Rourkela, Odisha, India
4 Infectious Diseases Biology Unit, Institute of Life Sciences, Bhubaneswar, Odisha, India
5 Department of Infection Biology, London School of Hygiene & Tropical Medicine, London, UK
*Corresponding author. Tel: +1 (646) 5016997; Fax: +1 (646) 5074645; E-mail: ana.rodriguez@nyumc.org

implicate XO as a host-derived source of inflammation in malaria. Accordingly, we observed that patients with elevated XO activity also present high levels of plasma inflammatory cytokines and higher incidence of cerebral malaria. We have also observed that *P. falciparum* acts as signal 2 and synergizes with ROS which act as signal 1 for the activation of the NLRP3 inflammasome and the release of IL-1β by macrophages.

## Results

To study the inflammatory response induced by *P. falciparum*, we incubated iRBC in the late developmental schizont stage with human macrophages derived from monocytes isolated from healthy donors. Little to no inflammatory cytokines are produced by these macrophages, which otherwise respond strongly to LPS (Fig 1A) and are fully viable (Appendix Fig S1). Similar results were observed when infected red blood cell lysates (iRBCL), which presumably contain potential pathogen-associated molecular patterns (PAMPs; Gazzinelli *et al*, 2014), were incubated with the macrophages (Fig 1B). A wide range of ratios of macrophages to iRBCL was tested in two independent experiments finding no detectable cytokine secretion at any concentration (Appendix Fig S2). Priming of macrophages with IFNγ (Fig 1C) or co-incubation with IFNα (Appendix Fig S3) before addition of iRBC (Glass & Natoli, 2015) also does not result in inflammatory cytokine secretion. We also observed that macrophages efficiently phagocytose iRBC (Fig 1D) and that incubation of macrophages with iRBC does not inhibit subsequent activation by LPS (Fig 1E), suggesting that the iRBC do not induce permanent inhibition of macrophage activation. Altogether, these data demonstrate that *P. falciparum* iRBC do not induce secretion of inflammatory cytokines by human monocyte-derived macrophages *in vitro*.

Malaria induces a "cytokine storm" in patients, where elevated levels of cytokines are found in the bloodstream (Clark, 2007; Clark *et al*, 2008). This is in contrast with our observations that *P. falciparum* does not induce even a minimal inflammatory cytokine response *in vitro* and suggests that other factors must be involved in triggering the host inflammatory response.

To study whether ROS, which are produced in large amounts during malaria, could contribute to the activation of the immune cells, we utilized XO, an enzyme involved in the oxidation of hypoxanthine and a potent producer of ROS (Battelli *et al*, 2016). XO is upregulated in children with malaria, and its levels increase with disease severity (Iwalokun *et al*, 2006). We observed that incubation of human macrophages *in vitro* with XO at a typical concentration found in the plasma of patients infected with *P. falciparum* (0.12 U/ml; Iwalokun *et al*, 2006) results in an increase of inflammatory cytokine secretion comparable to the one induced by LPS (Fig 2A). A dose–response analysis showed that increasing the concentrations of XO results in increased cytokine secretion by macrophages (Appendix Fig S4).

Increasing the concentration of hypoxanthine, the substrate for XO, results in higher levels of cytokine secretion (Fig 2B), while incubation with the specific inhibitor of XO febuxostat (Osada *et al*, 1993) results in decreased cytokine release (Fig 2C). These results confirm the XO specificity of the observed response and exclude the possibility that the cytokine secretion was caused by contamination

of the XO. Furthermore, inactivation of XO activity by heat treatment and addition of anti-oxidants, either membrane permeant or not, decreases cytokine secretion by macrophages incubated with XO (Fig 2D), indicating that the inflammatory activity is mediated by XO-produced ROS.

We also studied the response of whole peripheral blood mononuclear cells (PBMCs) to *P. falciparum* iRBC, whole or lysate, finding a weak cytokine response which was not significantly different than the RBC controls. PBMC also responded strongly to XO stimulation (Fig 2E).

XO produces ROS in the form of superoxide, which rapidly dismutates to $H_2O_2$ (Knowles *et al*, 1969) and permeates cell membranes reaching the cytosol (Bienert *et al*, 2007). We observed that cytokine secretion was not induced by the addition of $H_2O_2$ (Appendix Fig S5), which is short-lived within cells (Rhee *et al*, 2005). However, incubation of macrophages with XO for different times revealed that cytokine secretion requires exposure to the enzyme for at least 15 min (Fig 3A), suggesting that a prolonged exposure to ROS is necessary for the activation of the inflammatory response.

Since intracellular ROS are a well-characterized second messenger in the signaling cascade for immune cell activation (Schieber & Chandel, 2014), we next determined whether cytokine secretion could be triggered strictly by extracellular ROS or if it required intracellular production of these reactive molecules. We observed that physical separation of XO from macrophages through a dialysis membrane did not interfere with macrophage activation, indicating that extracellular ROS can induce cytokine secretion in macrophages (Fig 3B). These results expand the traditional view where activation of toll-like receptors (TLRs) or cytokine receptors induce intracellular ROS that act as second messengers, resulting in NFkB-mediated cytokine secretion (Nathan & Cunningham-Bussel, 2013).

To study the relevance of XO-produced ROS during *P. falciparum* infection, we first determined the relation between oxidative stress resulting from XO activity in the plasma of malaria patients and the levels of inflammatory cytokines. XO activity was measured by detecting ROS produced by plasma XO after addition of excess concentrations of its substrate, hypoxanthine. Under these conditions, the assay reflects the levels of oxidative stress produced by XO that are not quenched by anti-oxidants in the plasma, providing a measurement of the oxidant/anti-oxidant balance for each patient sample. We observed that two cytokines, TNF and IL-8, showed a significant correlation with plasma XO activity (Fig 4A), suggesting an involvement of this enzyme in malaria-induced inflammation. We also observed that patients with cerebral malaria showed higher levels of XO activity compared to patients with uncomplicated malaria (Fig 4B), suggesting a role for oxidative stress in the pathogenesis of this complication.

To determine whether plasma XO in malaria patients is responsible for triggering an inflammatory response in macrophages, we incubated plasma from malaria patients with macrophages *in vitro*. We observed that plasma samples from malaria patients can induce inflammatory cytokine secretion in macrophages that is inhibited by febuxostat (Fig 4C), indicating that XO is required for the generation of this response. Interestingly, the sample with higher levels of XO activity (twofold increase over healthy control plasma samples) corresponded to a patient with cerebral malaria and also induced a higher inflammatory response in macrophages. The plasma from

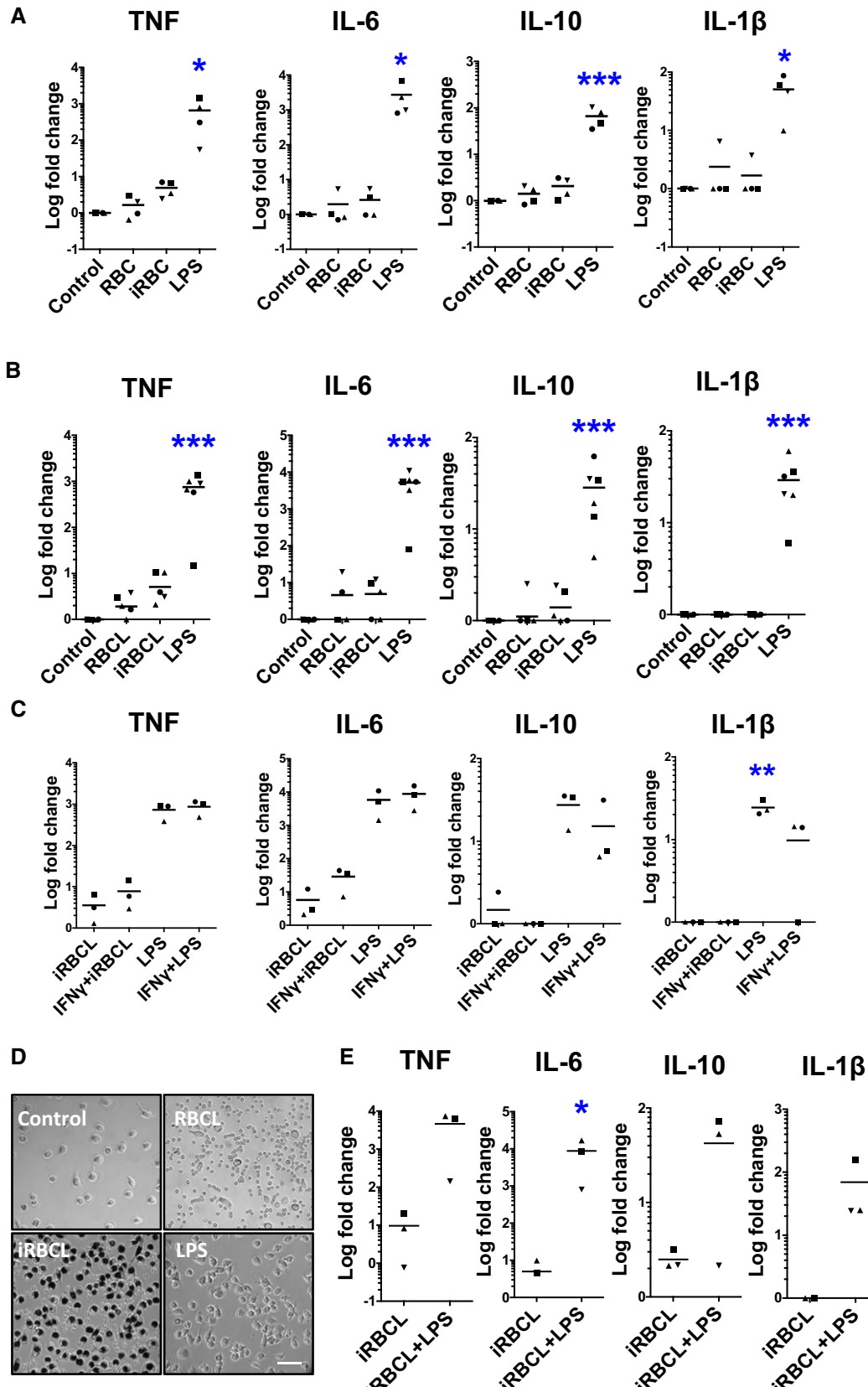

**Figure 1.**

◀

**Figure 1.** *Plasmodium falciparum*-infected erythrocytes do not induce inflammatory cytokine secretion in human monocyte-derived macrophages.

A, B  Macrophages produce little to no inflammatory cytokines when incubated with either (A) whole iRBC or (B) fresh iRBCL for 24 h. One-way ANOVA with Tukey test for multiple comparisons was performed to determine statistical significance.

C  Macrophages were pre-incubated with IFNγ (1,000 U/ml) for 18 h before iRBCL stimulation. One-way ANOVA with Tukey test for multiple comparisons was performed to determine statistical significance.

D  Phase-contrast microscopy of macrophages 24 h after the addition of RBC, iRBC, or LPS. Dark hemozoin crystals can be observed within macrophages. Scale bar is 30 μm.

E  Macrophages were incubated with iRBCL 24 h before addition of LPS. Paired *t*-test was performed to determine statistical significance.

Data information: Each symbol represents the value obtained for cells from an independent donor in an independent experiment. (A) $n = 4$; (B) $n = 5$; (C) $n = 3$; (E) $n = 3$. Asterisks indicate significance (*$P < 0.05$, **$P < 0.01$, and ***$P < 0.001$) when values were compared with RBC (A–C) or RBCL (E).

Source data are available online for this figure.

---

the two patients with uncomplicated malaria presented lower levels of XO activity and inflammation *in vitro*. It is important to note that no parasites or infected cells were added to this assay, suggesting that at least part of the inflammatory response triggered during malaria is not induced directly by the parasite.

Next, we investigated how oxidative stress contributes to inflammation in the context of malaria, where macrophages are simultaneously exposed to ROS and *Plasmodium*. We observed that when macrophages are incubated with *P. falciparum* iRBCL in the presence of XO, a synergistic increase is observed in the secretion of IL-1β, but not of TNF, IL-6, or IL-10 (Fig 5A). Although iRBCL alone did not induce secretion of IL-1β, it increased the secretion of this cytokine in response to XO by 20-fold. This increase is specific of iRBCL and dependent on its concentration (Fig 5B), indicating that infected, but not uninfected, erythrocytes synergize with ROS leading to increased release of IL-1β. Co-incubation of macrophages with XO and iRBCL also induced synergistic production of the chemokines IL-8, CCL5, and CCL2 (Fig 5C), which are essential in the recruitment and activation of leukocytes to sites of inflammation (Luster, 1998) and are also elevated during malaria (Ioannidis *et al*, 2014).

Elevated levels of IL-1β lead to inflammatory events such as fever, increased circulating neutrophils, upregulation of surface markers in cells, extensive nitric oxide production, and elevated levels of CRP and inflammatory cytokines (Dinarello, 1996, 2009; Cahill & Rogers, 2008), which are all common in malaria (Mackintosh *et al*, 2004). We propose that elevated levels of IL-1β induced by ROS contribute to the systemic inflammation in malaria.

The expression and release of IL-1β are mediated through the standard two-checkpoint model of priming and activation (He *et al*, 2016). Priming, considered signal 1, is classically mediated by the activation of TLRs, whose downstream signaling leads to the translocation of NFκB and the production of pro-IL-1β. Activation, or signal 2, promotes the assembly of a multimeric protein complex called the inflammasome, culminating in the autocatalyzation of caspase-1, the subsequent cleavage of pro-IL-1β, and the release of

bioactive IL-1β (Broz & Dixit, 2016). Gene expression analysis of macrophages incubated with XO shows greatly augmented IL-1β transcripts that are not much further increased when iRBCL are added (Fig 6A). Detection of pro-IL-1β protein in macrophages reveals that its presence is dependent on XO, but not on iRBCL (Fig 6B), indicating that XO is acting as signal 1 in macrophages. In contrast, activation of caspase-1 in macrophages is dependent on iRBCL, but not on XO (Fig 6C), assigning the function of signal 2 to iRBCL. This model is consistent with the synergistic increase in IL-1β secretion that is observed after the addition of iRBCL to XO-treated macrophages (Fig 5A and B). These data indicate that *priming* of macrophages is achieved by exposure to extracellular ROS, which promotes the production of pro-IL-1β, while the presence of infected erythrocytes mobilizes the inflammasome, propelling to the *activation* of caspase-1 and the consequent cleavage and release of bioactive IL-1β.

NLRP3 has consistently been implicated as a sensor for inflammasome activation in monocytes from both *P. falciparum* and *Plasmodium vivax* malaria patients (Ataide *et al*, 2014; Hirako *et al*, 2015). To determine whether NLRP3 is the inflammasome sensor molecule activated by *P. falciparum* in macrophages, we transfected human monocyte-derived macrophages with siRNA to knockdown expression of NLRP3. We achieved a 50% reduction in NLRP3 mRNA expression (Appendix Fig S6). When NLRP3-siRNA-transfected macrophages were incubated with both iRBCL and XO, they fail to secrete IL-1β compared to cells transfected with control siRNA (Fig 6D), implicating NLRP3 as the sensor that recognizes the parasite and leads the formation of the inflammasome in human macrophages. Conversely, the levels of other inflammatory cytokines and chemokines were not changed by NLRP3-siRNA transfection (Appendix Fig S7), which suggests that the increase observed in response to XO and iRBC stimulation (Fig 5C) is independent of IL-1β. Taken together, these results indicate that in human macrophages, extracellular ROS function as a signal 1, triggering the production of pro-IL-1β, while *Plasmodium* provides signal 2 for the activation of the NLRP3

---

**Figure 2.** Exposure to ROS promotes cytokine secretion from human macrophages and from PBMC.

A–E  Cytokine secretion of macrophages (A–D) or PBMC (E) incubated with the indicated stimuli for 24 h. RBC, iRBC, RBCL, iRBCL, XO, hypoxanthine (Hyx), the antioxidants *N*-acetyl-ʟ-cysteine (NAC), and 1-thioglycerol (1-TG), and heat inactivated (HI) XO were added to macrophages for 24 h. Febuxostat (Feb) was pre-incubated for 30 min with XO before addition to macrophages. Each symbol represents the value obtained for cells from an independent donor in an independent experiment. Insets show paired samples by donor in two different conditions. Blue asterisks indicate significance when values were compared with RBCL (A, E) or XO (B–D). Gray asterisks mark comparison with control (B–D). (A) $n = 7$; (B–E) $n = 3$. One-way ANOVA with Tukey test for multiple comparisons was performed to determine statistical significance (*$P < 0.05$, **$P < 0.01$, ***$P < 0.001$, and ****$P < 0.0001$).

Source data are available online for this figure.

▶

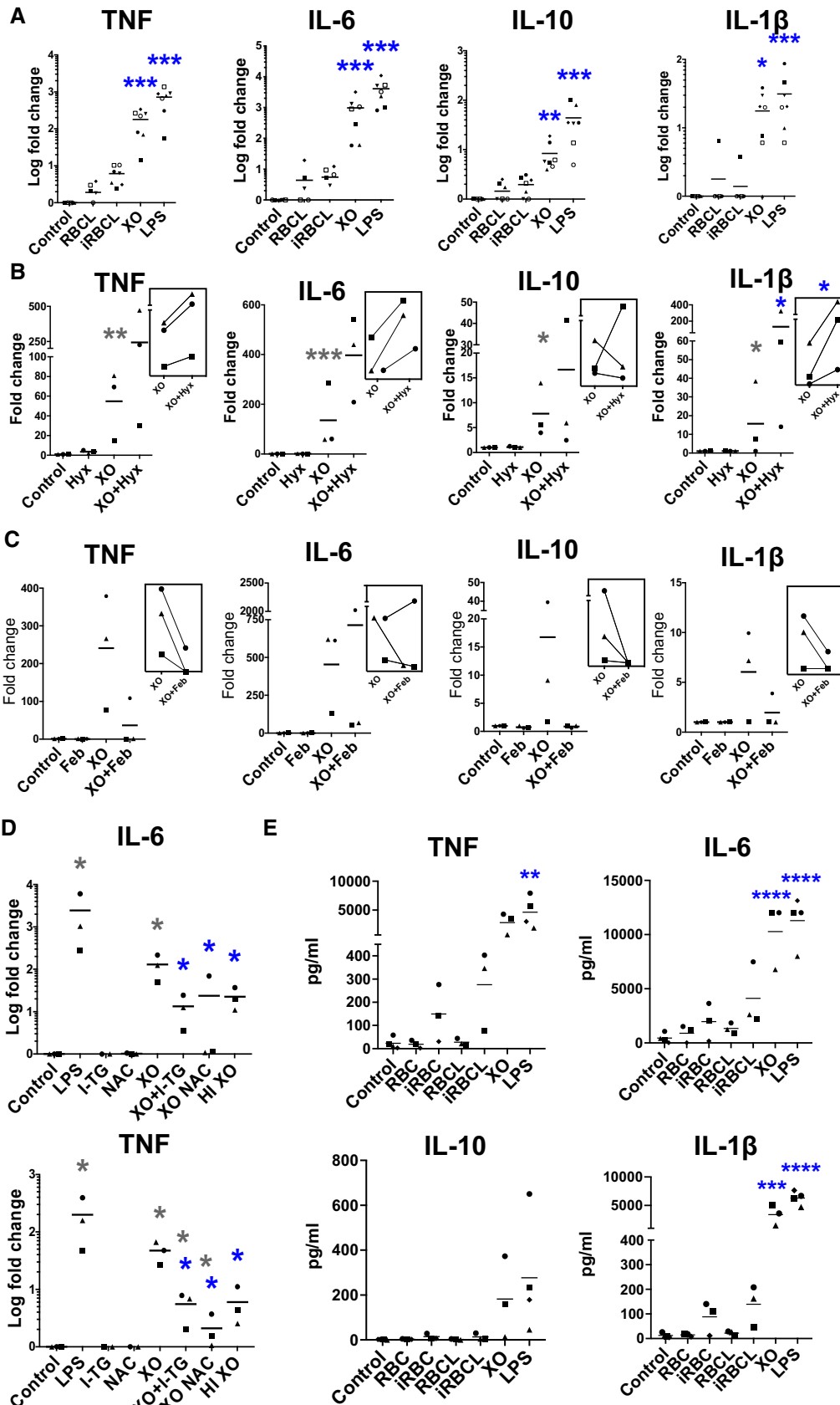

**Figure 2.**

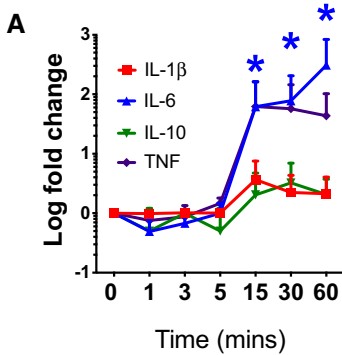

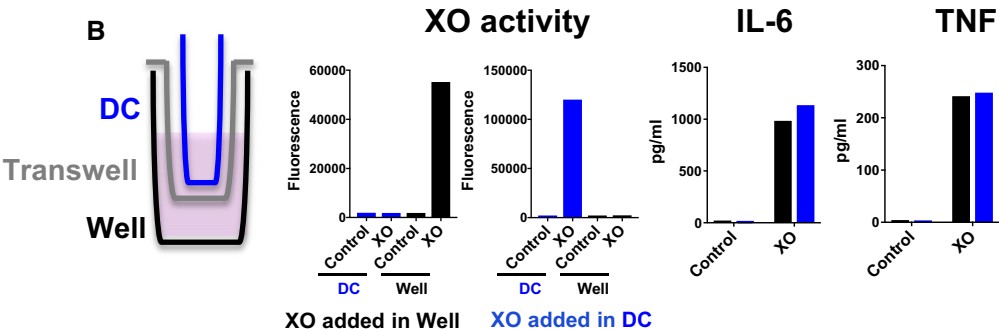

**Figure 3. Prolonged exposure to extracellular ROS induces cytokine secretion.**

A  XO was incubated with macrophages for the indicated times, when wells were washed twice and media was replaced. Supernatants were collected for cytokine determinations 24 h later (n = 3). Error bars show standard deviation. Kruskal–Wallis statistical method was used for statistical analysis. Blue asterisks indicate significance when values were compared with time 0 (*P < 0.05).

B  Diagram shows experimental set up with macrophages in the bottom of the well and XO added either in the dialysis cassette (DC, blue) or the well (black). XO activity panels show that the enzyme did not leak through the DC membrane. Secretion of IL-6 and TNF was measured 24 h after addition of XO in each compartment. A representative result of two independent experiments is shown.

Source data are available online for this figure.

inflammasome, resulting in the activation of caspase-1, which cleaves pro-IL-1β into mature IL-1β (Fig 6E).

## Discussion

Since the host inflammatory response contributes decisively to the pathology caused by *Plasmodium* infection (Mackintosh *et al*, 2004), understanding the mechanisms causing inflammation in malaria is of crucial importance. However, progress has been hindered by the lack of an *in vitro* model of inflammation since incubation of the parasite with macrophages or dendritic cells results in little to no response (Scragg *et al*, 1999; Elliott *et al*, 2007; Couper *et al*, 2010; Giusti *et al*, 2011; Zhou *et al*, 2012). Although PBMC incubation with iRBC results in detectable cytokine responses, which have been previously attributed mainly to monocytes and γδ-T cells (D'Ombrain *et al*, 2008; Stanisic *et al*, 2014), we found that they were not statistically significant and much lower than the responses observed to LPS. Several pathogen-associated molecular patterns have been identified in *P. falciparum*, including

**Figure 4. XO inflammatory activity in malaria patients and relation to cerebral malaria.**

A  Plasma levels of TNF and IL-8 from patients with uncomplicated (black circles, n = 14) or cerebral (red circles, n = 9) malaria correlate with levels of XO-produced ROS detected in each sample. ROS is expressed as fold change over plasma from healthy controls. Uncomplicated malaria, gray circles, cerebral malaria, red circles.

B  Levels of XO activity in malaria patients correlate with disease severity UM (uncomplicated malaria, n = 14) and CM (cerebral malaria, n = 9). Error bars show standard deviation.

C  Plasma from malaria patients induces macrophages to secrete cytokines, which are inhibited by a XO-specific inhibitor. Plasma from a healthy control (HP) and from three patients: P1 (with cerebral malaria), P2, and P3 (both with uncomplicated malaria), was pre-incubated for 30 min with febuxostat (Feb) or alone at 37°C before addition to macrophages at 1:2 dilution in media for 30 min. Cells were washed and cytokine secretion measured in triplicates after 24 h of incubation.

Data information: Linear regression (A), Mann–Whitney test (B), and unpaired *t*-tests (C) were performed to determine statistical significance (*P < 0.05, **P < 0.01).
Source data are available online for this figure.

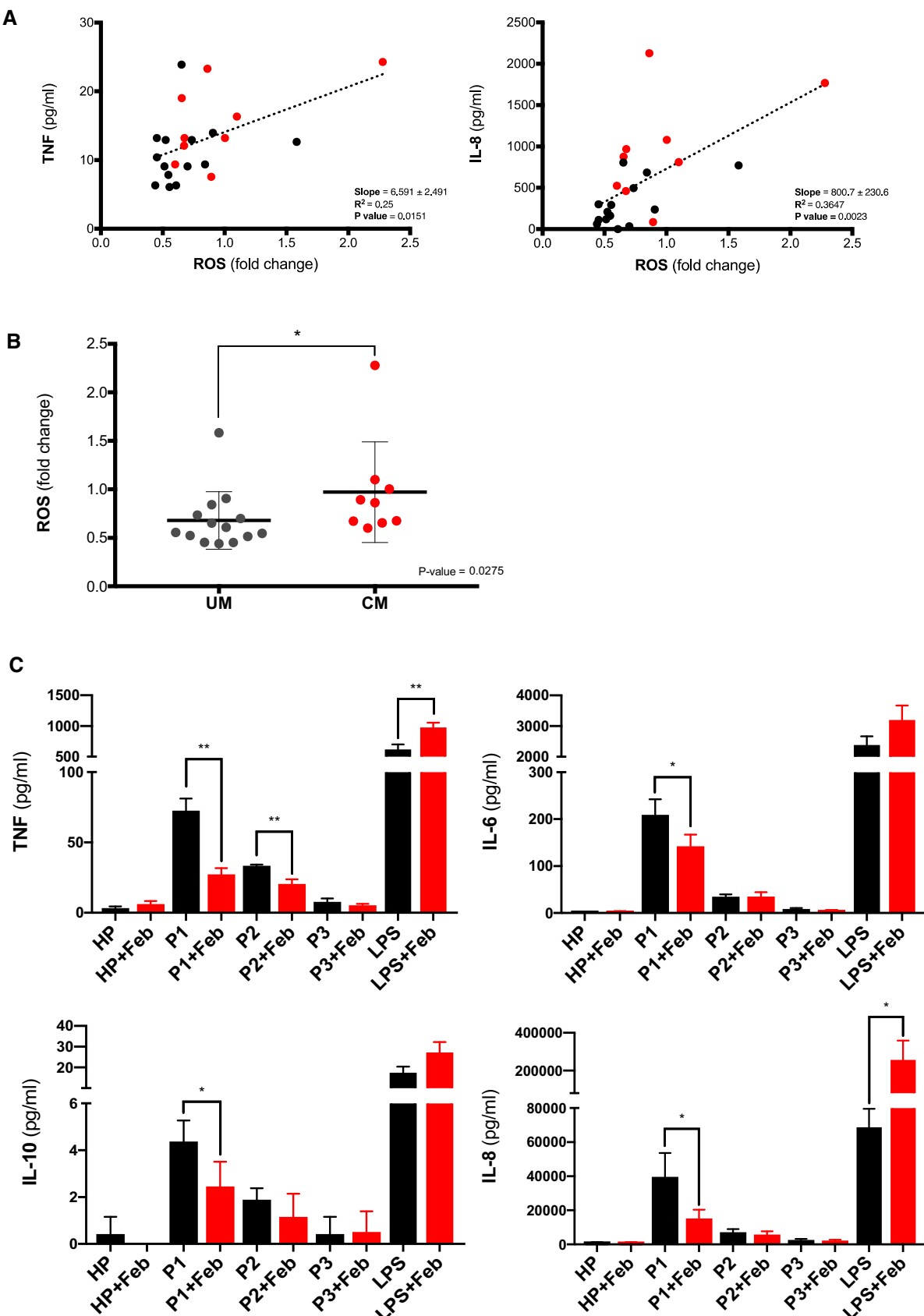

**Figure 4.**

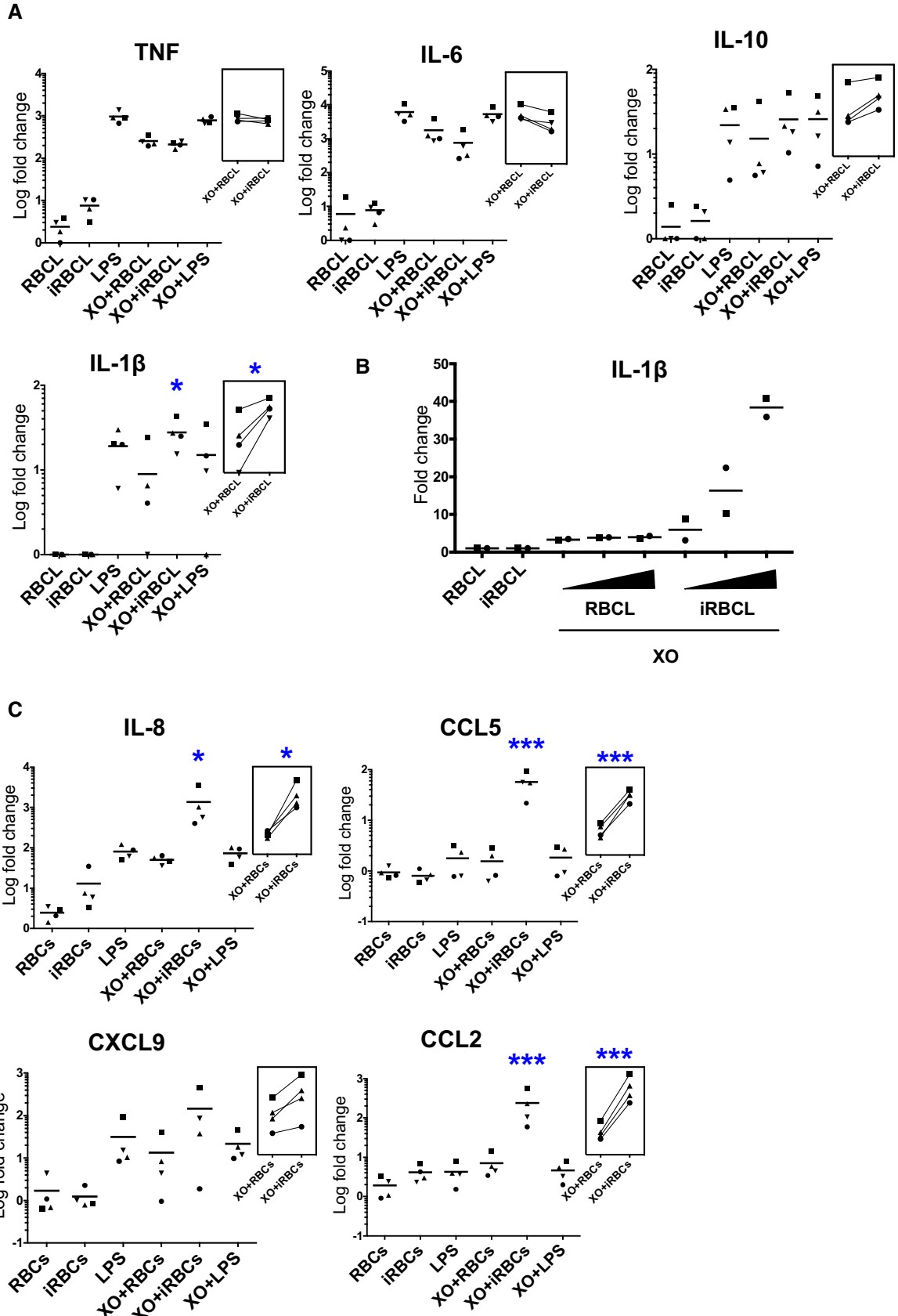

**Figure 5.**

**Figure 5.  ROS and *Plasmodium falciparum*-infected erythrocytes generate synergistic secretion of IL-1β and chemokines.**

A–C  Cytokine and chemokine secretion by macrophages after 24 h incubation with the indicated stimuli. (B) IL-1β levels after addition of increasing concentration of RBCL or iRBCL (1:2, 1:4, and 1:8 macrophage:iRBCL). One-way ANOVA with Tukey test for multiple comparisons was performed to determine statistical significance. Asterisks indicate significance when values are compared with XO + RBCL (*$P < 0.05$ and ***$P < 0.001$). Each symbol represents the value obtained for cells from an independent donor in an independent experiment. (A) $n = 4$; (B) $n = 2$; (C) $n = 4$. Insets show paired samples by donor in two different conditions.

Source data are available online for this figure.

GPI anchors, hemozoin, and parasite DNA (Gazzinelli *et al*, 2014). When purified from iRBC, these molecules are able to induce inflammatory cytokine responses from immune cells *in vitro* (Gazzinelli *et al*, 2014); however, these *in vitro* assays frequently use high concentrations of purified molecules that do not correspond to physiological concentrations of parasite and a direct comparison of purified molecules and whole iRBC or iRBCL were not performed. It is also possible that whole parasites are able to inhibit specific responses by these active molecules. Their relative contribution to malaria-induced inflammation and pathology in patients remains unclear (Erdman *et al*, 2008).

Considering the strong inflammatory responses observed in malaria patients, other factors beyond the parasite and infected erythrocytes must be involved in triggering the host inflammatory response. Our results point to oxidative stress, and in particular, extracellular ROS, as a trigger of inflammation in malaria since they are abundantly produced during infection and, as we describe here, are potent inducers of inflammatory cytokines.

An important role for oxidative stress in malaria-induced inflammation *in vivo* was suggested by a clinical trial to test the effect of allopurinol, an inhibitor of XO, in malaria. When treated with allopurinol in addition to a classical anti-malarial drug, severe malaria patients did not show significant improvement and only a small decrease in parasite levels ($P < 0.02$). However, a much faster decrease in inflammation [measured as fever ($P < 0.0002$) and splenomegaly ($P < 0.0002$)] was reported when compared to patients treated with the anti-malarial alone (Sarma *et al*, 1998), indicating that inhibition of XO has a strong inhibitory effect in malaria inflammation.

Our results where high levels of XO activity correlate with inflammatory cytokines support the hypothesis that oxidation induced by XO is an important cause of inflammation in malaria patients. It should be noted that the levels of anti-oxidants in the plasma will also influence the levels of active ROS in patients and therefore the levels of inflammation. In our assay, the levels of XO were measured as ROS activity detected after incubation with XO substrate, hypoxanthine; therefore, the ROS-neutralizing effect of

the plasma anti-oxidants influences the final ROS levels detected in every sample.

Additionally, we observed that higher levels of XO activity are found in patients with cerebral malaria supporting the hypothesis that oxidative stress is involved in the pathogenesis of this complication (Becker *et al*, 2004).

Our results where the plasma of some malaria patients triggered febuxostat-sensitive inflammatory cytokine responses in macrophages suggest an important role for XO in the generation of the inflammatory response to malaria. Although a higher number of patient samples will be needed to confirm this trend, the inflammatory activity of plasma from malaria patients already points to an important contribution of host-derived XO in the generation of the cytokine response. It is well-known that oxidative stress is associated with severity in malaria (Das *et al*, 1991; Greve *et al*, 2000; Narsaria *et al*, 2012), an effect that may be mediated indirectly through the ROS-induced elevated inflammatory reaction described here and/or directly through detrimental effects of oxidative stress in the tissues (Hemmer *et al*, 2005; Taoufiq *et al*, 2006).

Together with the observation that in the absence of ROS, human monocyte-derived macrophages produce little to no cytokines when they encounter infected erythrocytes or their lysates, these findings may explain puzzling observations in asymptomatic malaria patients, where detectable, or even high levels of parasites do not induce any fever (Bousema *et al*, 2014). This suggests that ROS may be pivotal for the inflammatory activation of immune cells, with the parasite having an adjunctive role for the induction of IL-1β and fever.

In this scenario, the upregulation of XO by type I IFN early in infection (Guermonprez *et al*, 2013) would be critical to induce the inflammatory response through the production of extracellular ROS. Since oxidative stress is linked to malaria severity (Das *et al*, 1991; Greve *et al*, 2000; Narsaria *et al*, 2012), this may explain the improved clinical symptoms observed when IFNAR1 is deficient or blocked in mice (Haque *et al*, 2014) and the link of specific SNPs in *IFNAR1* to cerebral malaria resistance in patients (Aucan *et al*, 2003).

**Figure 6.  ROS prime macrophages for *Plasmodium falciparum*-infected erythrocytes activation of the NLRP3 inflammasome.**

A  Gene expression analysis of macrophages incubated with the indicated stimuli expressed as fold change over RBCL. Dotted red line indicates 2-fold increase in RNA expression.

B  Western Blot analysis of pro-IL-1β. Relative abundance is the average of three independent experiments with different donors. Error bars show standard deviation.

C  FLICA analysis of activated caspase-1. Results are representative of two independent experiments with different donors.

D  Two distinct siRNAs against NLRP3 and an irrelevant control siRNA were utilized to knock down RNA levels by lipofectamine transfection in macrophages that were later incubated with iRBCL, or RBCL as control, with or without XO, before quantification of secreted IL-1β. Results are representative of two independent experiments with different donors.

E  Diagram of macrophage inflammasome activation by XO and iRBC.

Source data are available online for this figure.

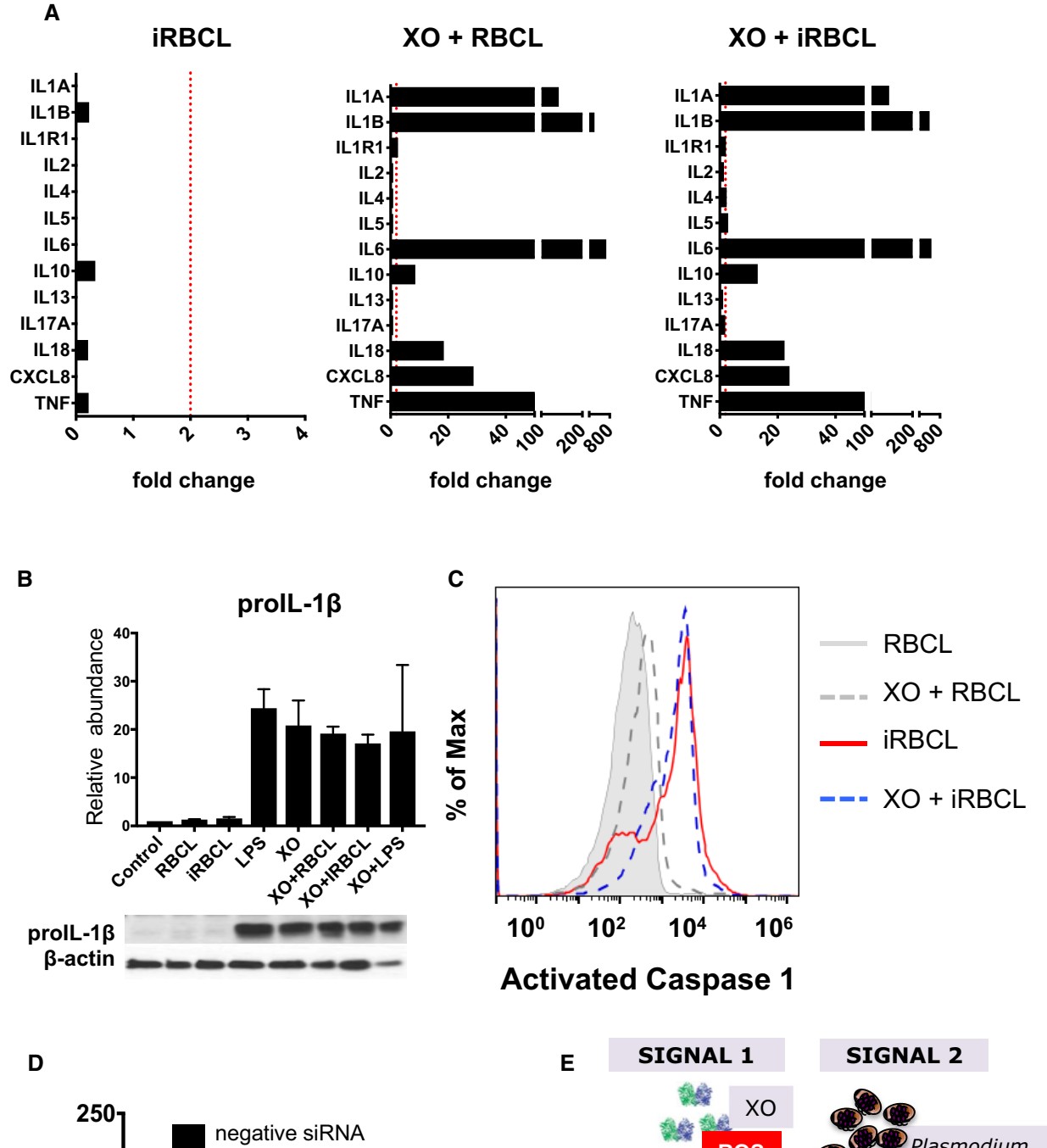

**Figure 6.**

In addition to elevated levels of host oxidative enzymes, such as XO, ROS in malaria may also be induced by the oxidative burst in macrophages that have phagocytosed iRBC (Ockenhouse *et al*, 1984; Kharazmi *et al*, 1987; Golenser *et al*, 1992) and the release of free heme from iRBC following synchronous schizont rupture (Andrade *et al*, 2010; Gozzelino *et al*, 2012). Although the relative contribution of these mechanisms in generating the high level of oxidative stress that is observed in malaria patients is not known, it is likely that they all contribute to the inflammatory response since they all result in the production of extracellular ROS. In particular, free heme has been implicated in the inflammatory response in malaria *in vivo* (Pamplona *et al*, 2007).

Previous reports have identified that *Plasmodium* infection activates the inflammasome in human patients (Ataide *et al*, 2014; Hirako *et al*, 2015); however, studies *in vitro* needed an additional exogenous signal 1 for the production of IL-1β that was added in the form of LPS, which is a bacterial product not present during malaria (Ataide *et al*, 2014; Kalantari *et al*, 2014; Hirako *et al*, 2015). Our results identify extracellular ROS, which are characteristic of malaria, as an effective signal 1 for the activation of the inflammasome in macrophages and provide a physiological explanation for those previous observations.

High levels of ROS are produced in a large number of infectious (Ivanov *et al*, 2017) and non-infectious diseases (Moylan & Reid, 2007; King, 2008; Reuter *et al*, 2010), but traditionally are considered intracellular messengers of inflammation, which are triggered in response to PAMPs, damage-associated molecular patterns (DAMPs), or cytokines (Nathan & Cunningham-Bussel, 2013; Broz & Dixit, 2016). During infections, intracellular ROS are generated by immune cells for killing of microbes through the oxidative burst (Warnatsch *et al*, 2017). Here, we propose a new role for extracellular ROS in directly inducing inflammatory cytokine secretion and synergizing with an infectious pathogen to activate the inflammasome in macrophages. Inhibition of this pathway through ROS inhibition or neutralization may have important implications for therapeutic use in diseases where inflammation triggered by oxidative stress contributes to the pathology.

## Materials and Methods

### Parasite culture and isolation

*Plasmodium falciparum* 3D7 asexual stage was cultured in RPMI 1640 with 25 mM HEPES, 25 mM sodium bicarbonate, and 1% gentamycin and enriched with 0.5% Albumax (pH 6.75) and 250 μM hypoxanthine. Parasitemias were measured every other day by counting Giemsa (Sigma) stained blood smears, and cultures were maintained at less than 5% parasitemia. Flasks were kept at 37°C, under atmospheric conditions (5% oxygen, 5% carbon dioxide, and 90% nitrogen). To retain synchronous cultures, parasite growth was treated with 5% D-sorbitol, which lyses late-stage iRBC leaving only ring stages. For experiments, schizont isolation was achieved using MACS cell separation LS columns (Miltenyi Biotec) and stored in −80°C. Mycoplasma contamination was assayed monthly using the MycoAlert Mycoplasma Detection Kit (Lonza) and found to be consistently negative.

### Parasite lysate preparation

As a control, RBC were processed in parallel with iRBC and kept in volumes of 100 μl or less. Quickly, 10 freeze–thaw cycles (5 s freeze: 1 min thaw) were performed between liquid nitrogen and the 37°C water bath. After the last cycle, the lysates were immediately added to the experimental plate.

### Human cell isolation and differentiation

Fresh whole blood was isolated from healthy donors at the NYU blood bank or Clinical and Translation Institute (CTSI) of NYU School of Medicine in accordance with the guidelines established by the NYU School of Medicine Institutional Review Board (IRB). Sterile filtered 7.22% sodium citrate (Sigma) was mixed with 450 ml of peripheral venous blood kept in constant agitation to avoid coagulation. Isolation and enrichment of peripheral blood monocytes were performed by density gradient centrifugation utilizing 30% Ficoll-Paque PLUS (GE Life Sciences). Next, PBMCs were allowed to attach to tissue culture plates (Falcon) and washed after 2 h, so only strongly adherent monocytes were left. After an overnight incubation, isolated monocytes were plated in 24 wells at a concentration of 500,000 cells/well in DMEM (Corning) and 1% Penicillin, Streptomycin, and Glutamine mixture (PSG). Cells differentiated for 7–12 days in 10% Human Sera (Valley Biomedical).

### Macrophage and *Plasmodium falciparum* co-cultures

Macrophage experiments were performed in 24-well plates in DMEM (Corning) 10% human sera and 1% PSG in the presence or absence of 0.12 U/ml xanthine oxidase (from bovine milk; Sigma) and 1 μg/ml LPS (Sigma). RBCLs and iRBCL were added at a 1:8 ratio (macrophage: RBC/iRBC). Twenty-four hours after experiments, supernatants were obtained and stored at −80°C unless otherwise noted.

### XO heat inactivation

XO at a concentration of 1.2 U was mixed with 100 μl of DMEM, heat inactivated at 95°C for 5 min, and immediately added to wells to a final concentration of 0.12 U/ml.

### Measurement of XO activity

XO activity was measured using Amplex Red® Xanthine/Xanthine Oxidase Assay Kit (Life Technologies) to detect ROS formation following manufacturer's instructions. A plate reader (Victor X3; Perkin Elmer) was utilized for measuring reactions with fluorescent excitation at 530 nm and emission at 590 nm. Background values were subtracted during the analysis. XO production of ROS was confirmed for each batch of the commercial enzyme (Sigma). In the experiment with human samples, the average of determinations on three healthy control samples was used as negative control and reference for XO activity in plasma samples.

### Cytokine and chemokine measurements

Supernatants collected from experiments and plasma samples were stored at −80°C. Inflammatory cytokines (IL-1β, IL-10, IL-6, and TNF)

and chemokines (IL-8, CCL5, CXCL9, and CCL2) were quantified using Cytometric Bead Array (CBA, BD Biosciences) and performed on FACS Calibur (BD Biosciences) according to manufacturer's protocol. The limit of detection for each cytokine or chemokine was lower than 10 pg/ml. FCAP array (BD Biosciences) and FlowJo (TreeStar) were utilized for quantification and further analysis.

### Quantitative RT–PCR

Macrophages were isolated and differentiated as described above and then incubated with iRBCL at a ratio of 1:8 for 6 h. RNeasy® Plus Mini Kit (Qiagen) was used for RNA extraction, and RT$^2$ First Strand kit (Qiagen) was utilized for subsequent cDNA synthesis following manufacturer's instructions. Analysis for macrophage gene expression was analyzed using RT$^2$ Profiler PCR Array Human Innate and Adaptive Immune Responses (Qiagen).

### Western blot analysis

Human monocyte-derived macrophages were washed 2× with 1× PBS (Fisher) and stored in −80°C until further analysis. Frozen cell lysates were washed, and lysis buffer comprised of RIPA buffer with protease inhibitor; PMSF (Sigma) was added to the wells to obtain cell lysates. After which, cell lysates were equalized for loading into a gradient SDS–PAGE gels (Bio-Rad) using the BCA Protein Assay Kit (Pierce). Proteins were separated by Tris–Hcl gel electrophoresis (Bio-Rad) and transferred onto PVDF membranes (Bio-Rad). Later, membranes were blocked overnight with blocking buffer (5% milk in 1× PBS-T with 0.1% Tween-20). For primary incubation, antibodies against IL-1β and β-actin (Appendix Table S1) were incubated for an hour, followed by another 1-h incubation with HRP-coupled secondary antibody 1:2,000 (Cell Signaling). Membranes were developed using Amersham® ECL Western Blotting Detection Reagents (GE Healthcare) following manufacturer's instructions. Quantification of protein band density was performed using ImageJ (NIH) and normalized to β-actin.

### siRNA transfections and knockdowns

Cells were isolated and differentiated as described above without antibiotics. Once majority of cells are attached (70–90%) exhibiting roundish macrophage/myeloid morphology (around day 4–5), cells were deemed ready for transfection. Media was aspirated gently from each well and washed twice with warm DMEM to ensure removal of floating cells. Fresh DMEM was added to a final volume of 250 μl and kept at 37°C with 5% $CO_2$. While incubating, 3% vol/vol of HiPerfect transfection (Qiagen) was mixed with 200 nM of NLRP3 siRNA (HS CIAS1-6 and HS CIAS1-9, Qiagen) or negative control siRNA (AllStars Negative Control, Qiagen) and allowed to form complexes for 20 min at room temperature. After which, the complexes were added in a dropwise manner and kept without any serum at 37°C with 5% $CO_2$. After 6 h, 500 μl of DMEM with 10% human serum was added and incubated overnight. Depending on the donor, another round of transfection was repeated after 24 h. To test for the levels of NLRP3 mRNA, we performed quantitative RT–PCR using RNeasy® Plus Mini Kit (Qiagen) and primers (TGAAGAAAGATTACCGTAAGAAGTACAGA and GCGTTTGTTGAGGCTCACACT) for NLRP3 PCR amplification. The internal normalizer gene used was 18sRNA with primers

CAGCCACCCGAGATTGAGCA and GCGTTTGTTGAGGCTCACACT for PCR amplification.

### Patients

Patients with cerebral malaria or uncomplicated *P. falciparum* malaria were admitted to Ispat General Hospital in Rourkela, India, from January 2012 to March 2014, as described elsewhere (Fernandez-Arias *et al*, 2016). Blood samples were collected upon admission, and plasma was frozen at −80°C for storage. 10% of the plasma samples were shipped to New York University School of Medicine for analysis, in accordance with the rules and regulations of the Government of India. The parent study in India has approval for the use of human subjects from the Institutional Review Boards from the New York University School of Medicine, the London School of Hygiene and Tropical Medicine, and from the Ispat General Hospital, Rourkela, India. All studies involving human subjects were conducted in accordance with the guidelines of the World Medical Association's Declaration of Helsinki. All individuals and/or their legal guardians gave written informed consent. The clinical data and samples were de-identified using a unique study number, in accordance with the Health Insurance Portability and Accountability Act.

Patients include adults and children over the age of 5 of both genders. All patients considered positive for cerebral malaria satisfied a strict definition: coma (defined as Glasgow Coma Score > 9 out of 15 for adults and Blantyre Coma Score > 2 for children) after correction of hypoglycemia (< 2.2 mmol/l), presenting asexual forms of *P. falciparum* in a peripheral blood smear sample, with or without associated complications such as renal failure, severe anemia, and jaundice. Plasma from three patients with malaria was used for Fig 4C: P1: male, age 46, cerebral malaria, fatal; P2: male, age 52, uncomplicated malaria, survived; P3: male, age 24, uncomplicated malaria, survived. Patient's plasma was diluted to 50% with DMEM and either left untreated or pre-treated with 500 μM febuxostat (0.6% DMSO) for 30 min at 37°C, and the same concentration of DMSO was added to control wells. Macrophages were then incubated with patient's plasma for 30 min, after which, cells were washed twice, and fresh DMEM with 10% human sera and 1% PSG was added to cells. Supernatants were collected 24 h later for cytokine quantification.

### Statistical analysis

Statistical analysis of all data was accomplished using GraphPad Prism 7. D'Agostino-Pearson omnibus normality test was utilized to determine if data sets were normally distributed. Once normality was established, one-way ANOVA was used as the statistical method with Tukey test for multiple comparisons. For data that did not pass the test (not normally distributed), Mann–Whitney, Wilcoxon, or Friedman test paired with Dunn's test for multiple comparison was used. Data were considered significant if $P < 0.05$. Asterisks denote level of significance: *$P < 0.05$, **$P < 0.01$, ***$P < 0.001$. Linear regression analysis was used to analyze the relation of patient plasma cytokines and ROS.

**Expanded View** for this article is available online.

### Acknowledgements
We thank CTSI resources NIH/NCATS 3UL1 TR001445, as well as the Director in Charge and the clinical staff of Ispat General Hospital in Rourkela for their

**The paper explained**

**Problem**

Malaria is a highly inflammatory disease. While inflammation contributes to eliminate the causative parasite, *Plasmodium falciparum*, excessive and persistent inflammation also contributes to severe malaria pathology. Despite its high clinical relevance, the source of inflammation in malaria patients is still not well defined.

**Results**

Here, we characterize that oxidative stress in the form of reactive oxygen species (ROS) produced by xanthine oxidase, an enzyme upregulated during malaria, results in high macrophage activation and release of inflammatory mediators. In combination with *P. falciparum*-infected erythrocytes, ROS activate the inflammasome and induce IL-1β secretion by macrophages. Xanthine oxidase activity in the plasma of malaria patients can induce an inflammatory cytokine response in macrophages *in vitro*, suggesting an important clinical relevance for this process.

**Impact**

Our findings indicate that a host enzyme upregulated during malaria induces an inflammatory response in macrophages, suggesting a shift in the generalized view that the malaria inflammatory response is triggered solely by the parasite. Our findings also point to oxidation in general, and xanthine oxidase in particular, as a novel target for anti-inflammatory therapies in malaria.

help and dedication. M.C.T. was supported by training grant 5T32AI007180; A.G. was supported by a fellowship within the Postdoc-Programme of the German Academic Exchange Service (DAAD). M.Z. was supported by 3R01HL130630-03S1; S.C.W. was supported by U19AI089676 and 3R01HL130630-03S1. The content is solely the responsibility of the authors and does not necessarily represent the official views of the NIH.

## Author contributions

MCT performed the studies, analyzed data, and wrote the article. AG and MZ helped with most of the studies. AM and SM supervised the enrollment of patients and collection of blood samples; SK, PKS, SM, and SCW processed the samples, analyzed data, and helped with the manuscript. AR provided overall supervision, analyzed the data, and wrote the article.

## Conflict of interest

The authors declare that they have no conflict of interest.

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
