## [Review Process File · EMBO Molecular Medicine]

Malaria Inflammation by Xanthine Oxidase-produced Reactive Oxygen Species

Maureen C. Ty, Marisol Zuniga, Anton Götz, Sriti Kayal, Praveen K. Sahu, Akshaya Mohanty, Sanjib Mohanty, Samuel C. Wassmer and Ana Rodriguez.

Review timeline:

Submission date:	3 rd October 2018
Editorial Decision:	8 th November 2018
Revision received:	15 th April 2019
Editorial Decision:	16 th May 2019
Revision received:	29 th May 2019
Accept:	5 th June 2019

Editor: Celine Carret

Transaction Report:

1st Editorial Decision

8th November 2018

Thank you for the submission of your manuscript to EMBO Molecular Medicine. We have now heard back from the two referees whom we asked to evaluate your manuscript.

As you will see from the reports below, the referees find the topic of your study of potential interest. However, they both raise substantial concerns on your work, which should be convincingly addressed in a major revision of the present manuscript. We would like to particularly emphasize that although we would not ask you to provide a *P. berghei*/mouse model as suggested by ref.2, we still would insist on performing the experiments with whole blood as suggested by ref. 1. Pathophysiological significance is key for EMBO Molecular Medicine, and therefore these experiments are needed for further consideration of the article. In addition, we also would like to encourage you to address all other issues listed to improve conclusiveness and clarity.

REFeree REPORTS

Referee #1 (Comments on Novelty/Model System for Author):

The authors try to make the case with a very reductionist approach. Most of their cellular work relies on the use of monocyte-derived macrophages. Blood-stage malaria is a systemic infection, and as such, the iRBC interacts with different cell type in the blood and in tissue. And this is overall interacts which will lead to the pro-inflammatory immune responses observed in patients.

In order to improve significantly the manuscript, they would have to repeat the experiments with a more relevant system: i.e. using whole blood stimulation or PBMC

Referee #1 (Remarks for Author):

Ty et al. have studied the role of Xanthine-oxidase-produced ROS in malaria inflammation. This is an interesting and original study. However, there are many experimental weaknesses. In particular the use of only one single cell type, supposedly mimicking in vitro the in vivo situation

(where the parasite interact with many different cell types in the blood and in tissues) is not conclusive. This prevents to draw definitive conclusions. The authors should also be more critical with their data and discuss how they differ with those of previous published studies.

Specific comments

1. In the introduction, the authors used inflammation indiscriminately. Do they refer to a pro-inflammatory response mediated by the release of pro-inflammatory cytokines? Is it fever? They should be clearer.

2. Most of their cellular work relies on the use of monocyte-derived macrophages. Blood-stage malaria is a systemic infection, and as such, the iRBC interacts with monocytes and macrophages present in all tissues. There is a large body of literature describing monocytes heterogeneity. Only for circulating monocytes, there are 2 major subsets with different biological functions (Stansfield et al, Clin Trans Med, 2015, 4, 5) (and see next comment). Thus the authors cannot draw general conclusions.

Page 4: line 30-35: the authors claims that malaria iRBC do not induced inflammatory cytokine in vitro? Although this seems true in their experimental systems (fig 1), they may not be the case when using other cells. To prove their claim, they should extend their findings using peripheral monocytes, neutrophils or dendritic cells?

In addition, they try to make the case, using a single cell type. They have to remember, that many leukocyte subsets are present in the peripheral blood or in tissues and that these cells interact directly or through mediators (i.e. cytokines) (Corrigan and Rowe, Parasite Immunol, 2010, 32, 512). In particular, Gamma delta T cells have been shown to be important in releasing IFN-gamma, an important pro-inflammatory cytokines (D'Ombain et al, Clin Infect Dis, 2008; Stanisic et al, 2014, 201, 295).

3. Their data contradict previous studies which has that parasite molecules such as GPI (Krishnegowda et al, J Biol Chem, 2005, 8606 ; Zhu et al J Biol Chem, 2880, 8617); hemozoin or tyrosyl-TRNA can induce macrophages or to release pro-inflammatory cytokines. These studies are not mentioned or even discussed in view of their results.

4. What was the rationale to choose a 1:8 ratio (macrophages/iRBC. The authors should do a dose effect.

5. What is the detection limit of the CBA array for each cytokines? This should be added

6. A table should be provided with more information of the patients from which the sera originated; were there different in ages, sex,

7. Statistical analysis: please indicate in the figure legends, which test was used, since different tests were used in this manuscript.

8. A dose effect of XO should be performed

10. Figure 2e: the elevation was only true for IL-6 .This should be specified since the authors make the case after for IL1beta. When is IL-1 beta produced?

11. Figure 3 b and c are not conclusive. There is no statistical analysis provided.

12. Figure 4: The authors claimed that IL-8, CCL5 and CLL2 are induced by IL1 beta after stimulation. By XO and iRBC .To support this claim, they should perform an experiment using a neutralising anti-IL1beta antibody. The follow-up experiments are based on this assumption and if IL-1beta is not involved in the secretion of these mediators, their conclusions would be erroneous.

13. Figure 5 C. the differences seen by Flow cytometry are minimal. The authors should do a statistical analysis with more data point.

Figure 5 d: no statistical analysis is provided.

Referee #2 (Comments on Novelty/Model System for Author):

I have finished my review on the Manuscript titled "Malaria Inflammation by Xanthine Oxidase-produced Reactive Oxygen Species" by Zuniga et al. Even if I find the idea potentially interesting I have several concerns about the technical quality of the study that are stated in my comments to the authors. If they are able to address these concerns I will be happy to consider a recommendation for publication. Concerning Novelty, medical impact and adequacy of model system, please find them below

-Novelty: The study is novel and the idea is quite good but in my opinion something is missing to trigger the reader's enthusiasm.

-Medical impact: severe malaria is common and life threatening, the study here has potential but even if the author used an inhibitor of XO they don't argue anything about applications and they don't use a murine model of malaria to support their statements.

-Adequacy of model system: the authors should have supported their finding with a *in vivo* model (murine cerebral malaria model?)

Referee #2 (Remarks for Author):

In this study Zuniga and collaborators explored the source of inflammation in malaria disease and specially during the manifestation of cerebral malaria that is accompanied with a high inflammation. The authors state the fact that even malaria is a high inflammatory disease this inflammation cannot be reproduced *in vitro* and augmented that other factors besides the parasites themselves are responsible for the inflammation observed in severe malaria. Zuniga et al proposed a model to explain the inflammation where the synergistic effect between the Reactive Oxygen Species (ROS) produced by the enzyme Xanthine Oxidase (XO) and the parasites is responsible for the inflammation. They also showed that XO inhibition abolishes inflammation *in vitro*.

Nevertheless, there are several methodological and interpretative flaws that should be corrected/answered before to be published in this journal:

MAJORS CONCERNS

1. The whole study is based in the use of a lysate of *Plasmodium falciparum* infected/non-infected red blood cells, but if the parasite lysate prepared as indicated in material and methods section, is the supernatant or the pellet is not clear at all, this must be clarified.

2. Macrophage and *P. falciparum* co-cultures; the authors used a ratio of 1:8 (macrophages:RBC/iRBC) in their co-cultures. Apparently only the co-culture with either LPS or the recombinant XO have an effect in cytokine production. The problem here is that in Fig. 4, the authors decided to make a dose response experiment, incubating macrophages and RBC/iRBC at different ratios always in presence of XO. A control is missing here because the effect of the incubation without XO is not shown. That will be critical to support their statements.

3. XO use in culture: the authors don't state the concentration of LPS/endotoxin in the recombinant XO used. They should demonstrate that XO effects are not due to LPS contamination. That is essential to support their claims because only the LPS culture with macrophages induced a cytokine response when they are not incubated with XO. Also, the heat inactivation of XO (Fig 2d) does not seem to have a big effect making the LPS measurement essential. In addition, the LPS concentration used is not stated.

4. XO inhibition by Fluxostat: because of the potential implication in cerebral malaria treatment of XO by already existing and in use inhibitors the use of a vehicle control (water, DMSO???) is also essential. Moreover, the concentration of Fluxostat (500 μ M) seems to be high, making a dose effect study necessary.

5. ROS production by XO: if XO is producing ROS, this should be demonstrated even more if the production of ROS is defended in Fig 2e

6. Fig 3: this figure is difficult to understand; in the text only a reference to Fig 3 without the letter is made. Also, the authors state that ROS measured in Fig 3 a and b are produced by XO content in the patient plasma. Fig 3c uses only four samples that does not seem to be enough

7. Fig. 5a: Here, the authors stated that the fold change is over RBCL lysates but in the figure it seems that is over the iRBCL, that would mean that XO is independent of *Plasmodium* presence or not (that is the opposite that was shown in Fig 4b). That needs to be clarified.

Fig.5b: Western blot analyses are not very convincing to me, the authors showed only proIL-1 β , Caspase1 expression is necessary. Also the authors need to show the whole western blot to show the

non specific bands.

Fig 5c: The authors stated that XO is not enough to activated caspase 1 but this control is missing.

Fig 5d: the concentration of IL-1 β is not enough, authors should to show NLRP3 protein levels and Caspase 1 levels or activation.

8. Statistical analysis: beside the mention to statistical analysis in the material and methods section, the statistical analyses used, samples numbers, biological replicates are missing. In addition several figures (Fig 1 a,b,c,e; Fig 2 b,c,d; Fig 4 a,c) show statistical significance with only 3 or 4 cells (biological or technical replicates??) or donors, that not seems to be adequate to me.

MINOR CONCERNS:

1. Even if the authors use an inhibitor that is in use and is specific of XO, they don't comment on the potential therapeutic use of the mechanism described here.

2. I am wondering if the authors could use a model of cerebral malaria (for instance P.berghei/C57BL6 model) to add more interest to this study. Maybe using XO to treat mice and score cerebral malaria progression.

1st Revision - authors' response

15th April 2019

Response to reviewers:

Referee #1

The authors try to make the case with a very reductionist approach. Most of their cellular work relies on the use of monocyte-derived macrophages. Blood-stage malaria is a systemic infection, and as such, the iRBC interacts with different cell type in the blood and in tissue. And this is overall interacts which will lead to the pro-inflammatory immune responses observed in patients.

In order to improve significantly the manuscript, they would have to repeat the experiments with a more relevant system: i.e. using whole blood stimulation or PBMC.

The response to this comment is found below in point number 2.

Referee #1 (Remarks for Author):

Ty et al. have studied the role of Xanthine-oxidase-produced ROS in malaria inflammation. This is an interesting and original study. However, there are many experimental weaknesses. In particular the use of only one single cell type, supposedly mimicking in vitro the in vivo situation (where the parasite interact with many different cell types in the blood and in tissues) is not conclusive. This prevents to draw definitive conclusions. The authors should also be more critical with their data and discuss how they differ with those of previous published studies.

Specific comments

1. In the introduction, the authors used inflammation indiscriminately. Do they refer to a pro-inflammatory response mediated by the release of pro-inflammatory cytokines? Is it fever? They should be clearer.

The introduction has been modified to clarify this point.

2. Most of their cellular work relies on the use of monocyte-derived macrophages. Blood-stage malaria is a systemic infection, and as such, the iRBC interacts with monocytes and macrophages present in all tissues. There is a large body of literature describing monocytes heterogeneity. Only for circulating monocytes, there are 2 major subsets with different biological functions (Stansfield et al, Clin Trans Med, 2015, 4, 5) (and see next comment). Thus the authors cannot draw general conclusions.

Page 4: line 30-35: the authors claims that malaria iRBC do not induced inflammatory cytokine in vitro? Although this seems true in their experimental systems (fig 1), they may not be the case when using other cells. To prove their claim, they should extend their findings using peripheral monocytes, neutrophils or dendritic cells?

In addition, they try to make the case, using a single cell type. They have to remember, that many leukocyte subsets are present in the peripheral blood or in tissues and that these cells interact directly or through mediators (i.e. cytokines) (Corrigan and Rowe, Parasite Immunol, 2010, 32, 512). In particular, Gamma delta T cells have been shown to be important in releasing IFN-gamma,

an important pro-inflammatory cytokines (D'Ombra et al, Clin Infect Dis, 2008; Stanisic et al, 2014, 201, 295).

As requested by the reviewer, we have performed similar experiments using human PBMC from healthy donors and incubating them with P. falciparum infected erythrocytes (iRBC), control uninfected erythrocytes and LPS. Our results indicate that PBMC also respond weakly to iRBC. Although the response is slightly higher than in macrophages, the PBMC response to iRBC is not significantly different than the RBC controls and much lower than the one triggered by LPS. We also observed that addition of Xanthine Oxidase induces a strong inflammatory response from PBMC. These results are now included in Figure 2e.

We have been careful to limit the conclusions throughout the manuscript to macrophages. In the introduction and discussion sections, we describe the previous literature showing that incubation of macrophages or dendritic cells with Plasmodium results in little to no cytokine responses. In the case of stimulated PBMCs, the assays from D'Ombra and Stanisic show IFN- γ responses from gd T-cells. We have included a sentence in the discussion section to acknowledge these results citing these references.

3. Their data contradict previous studies which has that parasite molecules such as GPI (Krishnegowda et al, J Biol Chem, 2005, 8606 ; Zhu et al J Biol Chem, 2880, 8617); hemozoin or tyrosyl-TRNA can induce macrophages or to release pro-inflammatory cytokines. These studies are not mentioned or even discussed in view of their results.

A paragraph has been added in the discussion section to discuss this issue.

4. What was the rationale to choose a 1:8 ratio (macrophages/iRBC). The authors should do a dose effect.

Following the reviewer recommendation, we performed a dose effect of iRBC finding that doses of iRBC from 1:2 to 1:32 (macrophages/iRBC) did not induce cytokine secretion by macrophages. This is now included as supplementary figure 2.

5. What is the detection limit of the CBA array for each cytokines? This should be added

The limit of detection for each cytokine or chemokine was lower than 10 pg/ml. This is now stated in the methods section.

6. A table should be provided with more information of the patients from which the sera originated; were there different in ages, sex,

Patient information is now provided in the methods section.

7. Statistical analysis: please indicate in the figure legends, which test was used, since different tests were used in this manuscript.

The statistical analysis methods used have been added for each figure legend.

8. A dose effect of XO should be performed

Following reviewer's advice, we have now performed a new experiment to study the effect of XO dose on macrophages. We observed that increasing the concentration of XO resulted in increased cytokine secretion by macrophages. This is now shown in supplementary Fig. 4.

10. Figure 2e: the elevation was only true for IL-6. This should be specified since the authors make the case after for IL1beta. When is IL-1 beta produced?

The elevation was highly significant for IL-6 and TNF at 15, 30 and 60 min compared to time=0, however, the observed increase at 15 min of IL-10 and IL-1b was not significant. For IL-1 beta and IL-10 the intensity of the response was very variable depending on the donors, as it is shown in Fig. 2a-c.

11. Figure 3 b and c are not conclusive. There is no statistical analysis provided.

We have now performed the determinations of cytokines in triplicates and performed statistical analysis on the results. New Fig. 4b and c.

12. Figure 4: The authors claimed that IL-8, CCL5 and CLL2 are induced by IL1 beta after stimulation. By XO and iRBC .To support this claim, they should perform an experiment using a

neutralising anti-IL1beta antibody. The follow-up experiments are based on this assumption and if IL-1beta is not involved in the secretion of these mediators, their conclusions would be erroneous. *In our first submission, we stated that "...it is likely that the observed chemokine increase is mediated primarily by IL1 β ". To address whether IL-1 β mediates the secretion of the chemokines in our experimental system, we have analyzed the production of IL8, CXCL9, CCL5, and CCL2 in the knock down experiments (Fig. 6d), when IL-1 β is inhibited. We observed that the secretion of the chemokines was not altered in the absence of IL-1 β , indicating that this cytokine does not mediate the observed chemokine increase. It is important to note that none of the follow-up experiments were based on this assumption. We have now added supplementary figure 9 to address this point.*

13. Figure 5 C. the differences seen by Flow cytometry are minimal. The authors should do a statistical analysis with more data point.

Figure 5 d: no statistical analysis is provided.

We have now added the logarithmic scale to this graph (now Fig. 6c) to clearly show that the increase in Caspase 1 activation by iRBCL (+/- XO) compared to RBCL (+/- XO) is at least 10-fold. Experiment in Fig. 5d (now 6d) is representative of 2 independent experiments with different donors.

Referee #2 (Comments on Novelty/Model System for Author):

1. The whole study is based in the use a lysate of Plasmodium falciparum Infected/non-infected red blood cells, but if the parasite lysate prepared as indicate in material and methods section, is the supernatant or the pellet is not clear at all, this must be clarified.

The lysate was used as a whole, not spun to separate pellet and supernatant. This is now clarified in the methods text.

2. Macrophage and P. falciparum co-cultures; the authors used a ratio of 1:8 (macrophages:RBC/iRBC) in their co-cultures. Apparently only the co-culture with either LPS or the recombinant XO have an effect in cytokines production. The problem here is that in Fig. 4, the authors decided to make a dose response experiment, incubating macrophages and RBC/iRBC at different ratios always in presence of XO. A control is missing here because the effect of the incubation without XO is not show. That will be critical to support their statements.

Following reviewer's advice, we have now performed a new experiment to study the effect of XO dose on macrophages. We observed that increasing the concentration of XO resulted in increased cytokine secretion by macrophages. This is now shown in supplementary Fig. 4.

We have also performed a dose effect of iRBCL finding that doses of iRBCL from 1:2 to 1:32 (macrophages/iRBC) did not induce cytokine secretion by macrophages. This is now included as supplementary figure 2.

3. XO use in culture: the authors don't state the concentration of LPS/endotoxin in the recombinant XO used. They should demonstrate that XO effects is not du to LPS contamination. That is essential to support their claims because only the LPS culture with macrophages induced a cytokine response when they are not incubated with XO. Also, the heat inactivation of XO (Fig 2d) does not seem to have a big effect making the LPS measurement essential. In addition, the LPS concentration used is not stated.

The LPS concentration (1 μ g/ml) is now specified in methods section. We agree with the reviewer that it is fundamental to demonstrate that XO effects are not due to LPS (or other) contamination of XO. We have included a paragraph in the results section and details about the origin of XO in methods section to clarify that the effect of XO is not a result of contamination by LPS or other.

Several findings support this conclusion:

1. Figure 2c shows that addition of febuxostat inhibits the cytokine response to XO, which would not happen if LPS or other contaminant was the cause of the cytokine response.

2. Figure 2d shows that the cytokine response to XO is inhibited in the present of anti-oxidants (1-TG and NAC), which would not happen if contaminating LPS was the cause of the cytokine response.

3. Figure 2b shows that addition of hypoxanthine, the substrate of XO, increases cytokine production, which indicates that increased XO activity is increasing the production of cytokines.

4. XO was not produced by bacteria, but was isolated from bovine milk, making LPS contamination unlikely.

Regarding the effect of heat inactivation, we would like to point out that the level of cytokine production induced after inactivating XO by heat exposure (HI XO) is 9 to 10 fold lower when compared to active XO. This indicates that cytokine production was caused, at least in part, by a heat-sensitive agent (such as XO activity, but not LPS). It is likely that HI XO stimulated macrophages to a small extent because it forms aggregates upon precipitation induced by heat exposure. It is documented that aggregated proteins can induce inflammatory responses in macrophages (Ratanji et al., J. Immunotoxicol. 2014. 11(2):99-109).

4. XO inhibition by Fluxostat: because of the potential implication in cerebral malaria treatment of XO by already existing and in use inhibitor the use of a vehicle control (water, DMSO??? no stated neither) is also essential. Moreover, the concentration of Fluxostat (500 μ M) seems to be high, making a dose effect study necessary.

We have now clarified in the methods section that Febuxostat was dissolved in DMSO, which was added at the same concentration to control macrophages. The dose of Febuxostat was chosen based on a dose response where the maximal concentration of Febuxostat that was not toxic for macrophages was chosen.

5. ROS production by XO: if XO is producing ROS, this should to be demonstrated even more if the production of ROS is defended in Fig 2e

ROS production by XO was demonstrated directly in our laboratory every time a new batch of enzyme was purchased. We used Amplex Red® Xanthine/Xanthine Oxidase assay kit (Life Technologies), as detailed in the methods section.

6. Fig 3: this figure is difficult to understand; in the text only a reference to Fig 3 without the letter is made. Also, the authors state that ROS measured in Fig 3 a and b are produce by XO content in the patient plasma. Fig 3c uses only four samples that does not seem to be enough

We have now inserted references to fig. 4a, fig 4b and fig. 4c that were missing in the text. In Figs. 4a and b, ROS produced by XO were measured in the plasma of patients using Amplex Red® Xanthine/Xanthine Oxidase assay kit (Life Technologies) as detailed in the methods section. Due to the difficulties in obtaining and exporting severe malaria clinical samples outside of India, the volume and number of samples from patients was very limited.

7. Fig. 5a: Here, the authors stated that the fold change is over RBCL lysates but in the figure it seems that is over the iRBCL, that would mean that XO is independent of Plasmodium presence or not (that is the opposite that was shown in Fig 4b). That needs to be clarified.

The fold change is calculated over RBC lysates. The graph showing the effect of iRBCL shows low values because the effect of iRBCL on macrophages cytokine RNA expression is very low, similarly to secreted cytokine protein levels as detected in fig. 1.

Fig.5b: Western blot analyses are not very convincing to me, the authors showed only proIL-1 β , Caspase1 expression is necessary. Also the authors need to show the whole western blot to show the non specific bands.

Activated caspase-1 expression is shown in Fig. 6c. Gels showing all bands are included in supplementary Fig. 6.

Fig 5c: The authors stated that XO is not enough to activated caspase 1 but this control is missing.

We considered XO in the presence of RBCL the best negative control for the experiment since it is compared directly with XO in the presence of iRBCL. We observed that XO in the presence of RBCL does not activate caspase 1, therefore we conclude that XO is not sufficient to activate caspase 1.

Fig 5d: the concentration of IL-1 β is not enough, authors should to show NLRP3 protein levels and Caspase 1 levels or activation.

Caspase 1 levels of activation are shown in Fig. 6c. NLRP3 RNA levels are shown in the new supplementary Fig. 8.

8. Statistical analysis: beside the mention to statistical analysis in the material and methods section, the statistical analyses used, samples numbers, biological replicates are missing. In addition several

figures (Fig 1 a,b,c,e; Fig 2 b,c,d,; Fig 4 a,c) show statistical significance with only 3 or 4 cells (biological or technical replicates??) or donors, that not seems to be adequate to me.
Each symbol in the experiments in figs. 1, 2 and 4 represents an independent experiment performed with a different donor in a different day. Statistical analysis is detailed in methods section. The statistical test used for each experiment is detailed in the figure legends.

MINOR CONCERNS:

1. Even if the authors use an inhibitor that is in use and is specific of XO, they don't comment on the potential therapeutic use of the mechanism described here.
A sentence discussing the potential therapeutic use is now included in discussion section.
2. I am wondering if the authors could use a model of cerebral malaria (for instance P.berghei/C57BL6 model) to add more interest to this study. Maybe using XO to treat mice and score cerebral malaria progression.

2nd Editorial Decision

16th May 2019

Thank you for the submission of your revised manuscript to EMBO Molecular Medicine. We have now received the enclosed reports from the referees that were asked to re-assess it. As you will see the reviewers are now globally supportive and I am pleased to inform you that we will be able to accept your manuscript pending the following final amendments:

Please address the issues commented by referee 2. Please make sure to indicate everywhere exact n and describe clearly the number of biological and/or technical replicates as well as the number of times experiments where performed.

REFeree REPORTS.

Referee #1 (Comments on Novelty/Model System for Author):

The authors have extended in the revised manuscript their findings to PBMC.

Referee #1 (Remarks for Author):

The manuscript has been improved

Referee #2 (Remarks for Author):

The authors have made a considerable effort to address adequately to most of my concerns. However I must to express my concern about some points of the study.

-Statistical analysis: there is a general lack of explanations about the number of times experiments were repeated, for example:

Fig 1,2,5. If each symbol represents cells from an independent donor in a independent experiment, does it mean that only one technical replicate (well) per donor was made? Some of the Figures in those panels have only 4 symbols or even only 3, that does not seem enough to perform statistical analysis, at least each figure is representative of three independent experiments (biological replicates). In that case please state and also include the statistical analyse used.

Supplementary Fig 2: only two points are included, how can that be supportive of their statements?

Supplementary Fig 4: how many times what this done?

Fig.4c; only one donor is use for cerebral malaria (P1) and two for uncomplicated (P2,3) and authors still performed statistical analysis and make conclusions. Though I am agree that there is difference only one point (I assume because those are plain bar histograms) is not enough to perform statistics or make conclusion. A limitation in samples should not be a justification.

-Concentration of DMSO (Vehicle control): a concentration of 500 uM of DMSO seems very high an inappropriate. Did the authors means the equivalent concentration of DMSO use for a concentration of inhibitor of 500uM

-Fig Sup 8; can the authors show the protein levels of NLRP3 on the knockdown samples?

-Fig Sup 2 and 4: those figures are supposed to support the whole study. Nevertheless they are

expressed in concentration of cytokines and the main figures are in Fold change. One or other of the representation must to be use to support those findings.

-LPS effect: In Fig sup 2 and 4 the effect of LPS and is very low compared with the main figures

MINOR CONCERNS:

- Line 32: Please explain/change statement " malaria is a very oxidative disease.
- Please explain better Fig 6e
- By their conclusion the medical application is not clear
- WB: Fig Sup 6 a: are the upper bands the B-actin?
- In general the authors should be more rigorous with the way they write conclusions

2nd Revision - authors' response

29th May 2019

Response to requested modifications:

1) Please address the issues commented by referee 2. Please make sure to indicate everywhere exact n and describe clearly the number of biological and/or technical replicates as well as the number of times experiments where performed.

The number of biological and/or technical replicates and the n are included in the figure legends. Exact p values are in the Appendix figures.

3rd Editorial Decision

5th June 2019

We are pleased to inform you that your manuscript is accepted for publication and is now being sent to our publisher to be included in the next available issue of EMBO Molecular Medicine.

Corresponding Author Name: Ana Rodriguez

Manuscript Number: EMM-2018-09903-V2